# BERT Has More to Offer: BERT Layers Combination Yields Better Sentence Embeddings

**MohammadSaleh Hosseini**[*], **Munawara Saiyara Munia**[†], **Latifur Khan**[*]

[*]Department of Computer Science
[†]Department of Electrical and Computer Engineering
The University of Texas at Dallas
{seyyedmohammadsaleh.hosseini, munawarasaiyara.munia,
lkhan}@utdallas.edu

## Abstract

Obtaining sentence representations from BERT-based models as feature extractors is invaluable as it takes much less time to pre-compute a one-time representation of the data and then use it for the downstream tasks, rather than fine-tune the whole BERT. Most previous works acquire a sentence's representation by passing it to BERT and averaging its last layer. In this paper, we propose that the combination of certain layers of a BERT-based model rested on the data set and model can achieve substantially better results. We empirically show the effectiveness of our method for different BERT-based models on different tasks and data sets. Specifically, on seven standard semantic textual similarity data sets, we outperform the baseline BERT by improving the Spearman's correlation by up to 25.75% and on average 16.32% without any further training. We also achieved state-of-the-art results on eight transfer data sets by reducing the relative error by up to 37.41% and on average 17.92%. [1]

## 1 Introduction

Learning sentence vector representations is a crucial problem in natural language processing (NLP) and has been widely studied in the literature (Conneau et al., 2017; Cer et al., 2018; Li et al., 2020). Given a sentence, the goal is to acquire a vector that semantically and/or syntactically represents it. BERT (Devlin et al., 2019) has set new state-of-the-art records on many NLP tasks (Madabushi et al., 2020; Hu et al., 2022; Skorupa Parolin et al., 2022; Wang and Kuo, 2020). However, this is achieved by fine-tuning all of BERT's layers. The disadvantage of fine-tuning is that it is computationally expensive as even BERT-base has 110M parameters; hence, pre-computing a representation of the data and using it for the downstream task is much less computationally expensive (Devlin et al., 2019).

Moreover, for sentence-pair tasks, BERT uses a cross-encoder; nonetheless, this setup is inappropriate for certain pair regression tasks, such as finding the most similar sentence in a data set to a specific sentence due to the large number of possible combinations (Reimers and Gurevych, 2019).

Considering the aforementioned drawbacks, researchers have tried to derive fixed-sized sentence embeddings from BERT or proposed new BERTs with the exact same architecture but different ways of training (Reimers and Gurevych, 2019; Li et al., 2020). After training, the resultant BERT is used in a feature-based manner by passing the sentence to it and obtaining its embedding vector in different ways, such as averaging the last layer of BERT.

In this paper, we present a simple, yet effective and novel method called **BERT-LC** (BERT Layers Combination). BERT-LC combines certain layers of BERT in order to obtain the representation of a sentence. As we will show, this model significantly outperforms its correspondent BERT baseline with no need of any further training. Our work was inspired by Jawahar et al. (2019), who show that different layers of BERT carry different features, such as surface, syntactic, and semantic. We argue that each data set with its unique distribution might need a different set of features for its sentences, which can only be fully exploited by combining different layers of BERT in an unsupervised way.

Our contributions are as follows: (1) We propose a new method called BERT-LC that is capable of acquiring superior results by combining certain layers of BERT instead of just the last layer, in an unsupervised manner. We also include the embedding layer, which was to our knowledge ignored in previous works. (2) We additionally show that our method improves SBERT (Reimers and Gurevych, 2019) and SimCSE (Gao et al., 2021), which were specifically designed for obtaining sentence representations (opposite to BERT and RoBERTa (Liu et al., 2019)). (3) We developed an algorithm that

---

[1]Our code is available at: https://github.com/DiamondRock/BERT-Layers-Combination

speeds up the process of finding the best layer combination (among $2^{13}$ layer combinations in base cases) by a factor of 189 times. (4) We propose an innovative method that integrates the layer combination method with the CLS pooling head, improving the performance metrics for certain models. (5) We achieve state-of-the-art performance on the transfer tasks using layer combination.

We demonstrate the superiority of our approach through conducting extensive experiments on seven standard semantic textual similarity (STS) data sets and eight transfer tasks. On the STS data sets, our method is able to outperform its corresponding baseline by up to 25.75% and on average 16.32% for BERT-large-uncased. We also achieve the state-of-the-art performances on transfer tasks, reducing the previous best model's relative error rate by an average of 17.92% and up to 37.41%.

## 2  Related Work

Learning sentence embeddings is a well-studied realm in NLP. There are mainly two methods used for this purpose: methods that use unlabeled data or labeled data. Although the latter use labeled data, the target data sets and tasks on which they are tested are different from the training data set and task. Early work on sentence embedding utilized the distributional hypothesis by predicting surrounding sentences of a sentence (Kiros et al., 2015; Hill et al., 2016; Logeswaran and Lee, 2018). Pagliardini et al. 2018 build on the idea of word2vec (Mikolov et al., 2013) using n-gram embeddings. Some researchers simply use the average of BERT's last layer embeddings as the sentence embedding (Reimers and Gurevych, 2019). Recently, contrastive learning has proven to be very powerful in many domains (Khorram et al., 2022; Munia et al., 2021; Hu et al., 2021); thus, some recent methods exploit contrastive learning ( Gao et al., 2021; Zhang et al., 2022; Yan et al., 2021) and utilize the same sentence by looking at it from multiple angles. For instance, the popular method SimCSE leverages different outputs of the same sentence from BERT's standard dropout.

There are some previous works (Ethayarajh, 2019; Jawahar et al., 2019; Bommasani et al., 2020) that investigate the impact of different BERT layers. However, they either investigate each layer individually or are restricted to considering a set of consecutive layers strictly starting from layer 1.

We apply our method to different variations of

BERT, RoBERTa, SBERT, and SimCSE. To the best of our knowledge, no similar work has been done that whether combines different and arbitrary layers of these models or further integrates layer combination and the CLS pooling head.

## 3  Approach

Given an input sentence $S = (s_1, s_2, \ldots, s_N)$, the goal of a sentence embedding model is to output a vector $E_S \in \mathbb{R}^d$ which carries the semantic and/or syntactic information of the sentence. In order to obtain a sentence embedding, we first pass the sentence to a BERT-like model, which outputs the tensor $H \in \mathbb{R}^{L' \times N \times d}$, in which $d$ is the dimension of the token vectors in each layer, and $L' = L+1$ is the number of layers (including Layer 0) in BERT. We then apply to this tensor a pooling function $p$, which can be $max$ or $mean$, but we choose mean as it yielded better results in our experiments. The pooling is done across all the tokens in all the desired layers set, $D$. For example, for mean pooling, $p$ is defined as:

$$ p(H, D) = \frac{1}{|D|} \sum_{l \in D} \frac{1}{N} \sum_{n=1}^{N} H_{l,n,:} \qquad (1) $$

where $H_{l,n,:} \in \mathbb{R}^d$ is the Transformer vector at layer number $l$ corresponding to the $n$th token.

Previous works usually set $D$ to $\{L\}$ or use CLS pooler, the output of the MLP layer attached to the last layer's first token ([CLS]) to obtain the sentence embedding. Using CLS pooler was shown to underperform last layer averaging (Reimers and Gurevych, 2019). Further, as we will empirically show, choosing $D = \{L\}$ leads to significant underperformance as well; hence, in this work, we iterate through all possible $D$s ($D \in \mathcal{P}(A) - \emptyset$, where $A = \{0, \ldots, L\}$), and choose the best-performing $D$ as our layers to pool from.

As iterating through all possible $D$s is very time-consuming, we propose an algorithm that speeds up the process of finding the best layer combination ($2^{13}$ layer combinations in base cases) by a factor of 189 times, which can be found in Appendix A.

We further propose an extension to our method, and that is exploiting the MLP head and layer combination simultaneously. The idea is to pass $p(H, D)$ in Eq. 1 to the MLP head and use the output of the MLP head as the new $p$. We spot that this method works better than merely using the layer combination on SimCSE. We conjecture that this is because SimCSE's MLP head was trained for learn-

ing sentence embeddings as opposed to the other methods, such as BERT, SBERT, and RoBERTa; hence, it carries important information to be utilized. Consequently, we use the two-step pipeline of layer combination and MLP for SimCSE models, and we propose that the two-step pipeline is utilized for any other BERT-based sentence embedding model whose MLP head has been trained for sentence embedding.

# 4 Experiments and Results

We carry out our experiments on two different tasks: transfer and STS tasks. In this section, we discuss the tranasfer tasks, while the STS tasks are discussed in Appendix B.

## 4.1 Data Sets and Evaluation Setup

For transfer tasks, we use eight data sets from the popular SentEval toolkit (Conneau and Kiela, 2018), which is used for assessing the quality of sentence embeddings: MR (Pang and Lee, 2005), CR (Hu and Liu, 2004), SUBJ (Pang and Lee, 2004), MPQA (Wiebe et al., 2005), SST and SSTM (binary and six-class Stanford Sentiment Treebanks)(Socher et al., 2013), TREC (Li and Roth, 2002), and MRPC (Dolan et al., 2004).

For each data set, we combine all of its data subsets and randomly split them into training-development (train-dev) and test data with ratios 85% and 15%. This *outer* cross-validation is done randomly for 10 times, and the average accuracy results on the test data are reported. We further utilize a 10-fold *inner* cross-validation on train-dev data set, with ratios 82% (train) and 18% (dev).

The sentence embeddings by our method or the baselines are used as feature vectors for a logistic regression classifier. Note that even though there exists a training data set, it is only utilized by the logistic regression classifier and not in our method.

## 4.2 Baselines and Hyper-parameters

We use the following baselines and apply our layer combination method to them: BERT base/large (un)cased (**BERT-(B/L)**$_{(un)cased}$), RoBERTA base/large (**RoBERTa-B/L**), SBERT and SRoBERTa base/large (**SBERT/SRoBERTa-B/L**), (un)supervised SimCSE BERT/RoBERTa base/large (**(Un)SupSimCSE-(B/R)(B/L)**), and **SupSimCSE-RB/RL**$_M$, which are SimCSE models trained with an additional MLM head.

For the base and large models, we consider layer combinations with up to four and three layers, respectively, as we did not spot much difference when combining more layers.

For the logistic regression, we use SAGA (Defazio et al., 2014) as the optimizer, 0.01 as the tolerance level, 200 as the maximum number of iterations, and 10 as the regularization parameter.

## 4.3 Results and Discussion

The results for our proposed method and the baselines on transfer tasks are shown in Table 1. The results are statistically significant (p-value < 0.01). Note that our methods are denoted by an **LC** at the end of them (e.g., RoBERTa-L-LC means RoBERTa-L when layer combination is used).

Since all models have accuracies close to 100% on most of the transfer data sets, we believe it is more reasonable to see how much improvement we are acquiring by considering the relative error reduction. Relative error for a model and its baseline with respective accuracies of $A_M$ and $A_{MB}$ is defined as $\frac{A_M - A_{MB}}{1 - A_{MB}}$. From Table 1, we observe that our method significantly outperforms its corresponding baselines for all the models on all the data sets. For example, RoBERTa-B-LC, SBERT-L-LC, and UnSupSimCSE-RL-LC improve on their baselines by reducing the relative errors by up to 36.19%, 45.93%, and 40.80% and on average 19.89%, 14.95%, and 27.31%, respectively.

Furthermore, we spot that our method is more effective on unsupervised versions of SimCSE models. For instance, the relative error improvement over UnSupSimCSE-RL is 27.31%, whereas it is 14.61% for SupSimCSE-RL. However, there is still important information to be extracted from all the layers of the supervised SimCSE models as well.

We see that even an unsupervised model such as RoBERTa-L when upgraded with LC can achieve 84.58%, beating SupSimCSE-RL, a supervised model which held the previous best average result (83.73%). RoBERTA-L-LC's average accuracy does not fall very short of the new state-of-the-art model, SupSimCSE-RL-LC (84.68% vs. 85.89%). Finally, we have improved the previous state-of-the-art models on all of the transfer data sets by reducing the relative (absolute) errors by up to 37.41% (3.67%) and on average 17.92% (1.90%), respectively. We achieve a state-of-the-art average accuracy of 86.36% on the transfer data sets. We have provided more baselines, which can be found in Appendix D.

| | MR | CR | SUBJ | MPQA | SSTM | TREC | MRPC | SST | Avg. |
|---|---|---|---|---|---|---|---|---|---|
| BERT-B$_{uncased}$-CLS | 80.16 | 83.17 | 93.97 | 84.35 | 46.66 | 74.74 | 70.87 | 85.01 | 77.37 |
| BERT-B$_{uncased}$-last avg | 81.19 | 86.17 | 95.07 | 88.10 | 47.53 | 85.76 | 73.66 | 87.08 | 80.57 |
| BERT-B$_{uncased}$-LC | **82.24** | **86.60** | **95.40** | **90.42** | **49.22** | **88.76** | **77.17** | **88.17** | **82.25** |
| BERT-B$_{cased}$-CLS | 77.88 | 83.07 | 92.33 | 85.29 | 45.19 | 69.97 | 70.64 | 83.45 | 75.98 |
| BERT-B$_{cased}$-last avg | 80.55 | 85.19 | 94.64 | 87.63 | 46.48 | 83.65 | 74.30 | 85.83 | 79.78 |
| BERT-B$_{cased}$-LC | **81.42** | **86.56** | **95.11** | **89.88** | **47.76** | **88.00** | **75.36** | **86.86** | **81.37** |
| RoBERTa-B-last avg | 82.58 | 84.69 | 94.55 | 85.83 | 50.26 | 81.90 | 72.51 | 87.28 | 79.95 |
| RoBERTa-B-LC | **85.89** | **90.23** | **95.68** | **89.24** | **52.39** | **86.92** | **74.85** | **89.65** | **83.11** |
| UnSupSimCSE-BB | 80.67 | 85.29 | 94.29 | 88.75 | 45.14 | 83.96 | 72.94 | 85.86 | 79.61 |
| UnSupSimCSE-BB-LC | **81.61** | **86.77** | **95.16** | **90.03** | **47.07** | **89.05** | **77.29** | **87.49** | **81.81** |
| UnSupSimCSE-RB | 80.85 | 85.36 | 92.40 | 86.70 | 47.25 | 76.13 | 72.74 | 85.67 | 78.39 |
| UnSupSimCSE-RB-LC | **84.00** | **88.99** | **94.69** | **88.84** | **50.50** | **86.02** | **77.59** | **88.33** | **82.37** |
| BERT-L$_{uncased}$-CLS | 83.22 | 85.64 | 93.03 | 90.75 | 46.73 | 69.45 | 69.10 | 84.01 | 76.49 |
| BERT-L$_{uncased}$-last avg | 83.88 | 88.29 | 95.52 | 86.51 | 49.88 | 83.81 | 71.49 | 88.48 | 80.98 |
| BERT-L$_{uncased}$-LC | **85.21** | **89.88** | **96.08** | **90.17** | **51.06** | **88.89** | **76.57** | **89.96** | **83.48** |
| BERT-L$_{cased}$-CLS | 81.65 | 82.79 | 91.43 | 83.49 | 45.76 | 64.23 | 70.02 | 84.83 | 75.53 |
| BERT-L$_{cased}$-last avg | 84.28 | 88.68 | 94.95 | 88.03 | 49.23 | 84.10 | 72.62 | 88.75 | 81.33 |
| BERT-L$_{cased}$-LC | **85.34** | **90.19** | **95.41** | **90.50** | **50.92** | **88.62** | **77.17** | **90.00** | **83.52** |
| RoBERTa-L-last avg | 84.30 | 85.22 | 94.93 | 87.25 | 50.59 | 81.75 | 67.24 | 89.41 | 80.09 |
| RoBERTa-L-LC | **88.04** | **91.68** | **96.51** | **91.11** | **54.11** | **88.06** | **75.01** | **92.08** | **84.58** |
| UnSupSimCSE-BL | **84.85** | 88.15 | 95.09 | 89.13 | 48.84 | 83.63 | 74.09 | 89.02 | 81.60 |
| UnSupSimCSE-BL-LC | **84.85** | **89.84** | **95.77** | **90.62** | **50.93** | **89.07** | **77.10** | **90.04** | **83.53** |
| UnSupSimCSE-RL | 82.29 | 86.24 | 92.77 | 88.12 | 45.98 | 82.02 | 73.91 | 88.08 | 79.93 |
| UnSupSimCSE-RL-LC | **86.75** | **91.01** | **95.72** | **90.85** | **52.83** | **87.93** | **79.13** | **91.57** | **84.47** |

(a) Unsupervised Models

| | MR | CR | SUBJ | MPQA | SSTM | TREC | MRPC | SST | Avg. |
|---|---|---|---|---|---|---|---|---|---|
| SBERT-B | 82.90 | 88.96 | 93.93 | 89.58 | 47.53 | 80.04 | 74.34 | 89.19 | 80.81 |
| SBERT-B-LC | **83.61** | **89.95** | **95.17** | **91.06** | **49.01** | **88.56** | **78.71** | **89.65** | **83.22** |
| SRoBERTa-B | 84.67 | 90.12 | 92.57 | 89.21 | 50.59 | 81.79 | 77.13 | 90.12 | 82.03 |
| SRoBERTa-B-LC | **85.76** | **91.75** | **94.80** | **90.51** | **53.65** | **87.95** | **78.92** | **90.78** | **84.26** |
| SupSimCSE-BB | 81.85 | 89.31 | 94.60 | 89.73 | **50.16** | 82.75 | 74.60 | 88.81 | 81.48 |
| SupSimCSE-BB-LC | 81.85 | 89.31 | **95.68** | **90.87** | **50.16** | **89.09** | **77.68** | **89.67** | **83.04** |
| SupSimCSE-RB | 84.29 | 91.50 | 93.12 | 90.19 | 52.69 | 81.12 | 76.14 | 90.14 | 82.40 |
| SupSimCSE-RB-LC | **86.00** | **92.38** | **95.21** | **90.89** | **54.00** | **87.97** | **78.76** | **90.97** | **84.52** |
| SupSimCSE-RB$_M$ | 84.56 | 91.89 | 93.29 | 89.49 | 52.28 | 81.88 | 76.23 | 90.13 | 82.47 |
| SupSimCSE-RB$_M$-LC | **86.11** | **92.38** | **95.81** | **90.63** | **54.10** | **88.40** | **78.14** | **91.00** | **84.57** |
| SBERT-L | 84.69 | 90.62 | 94.40 | 90.25 | 49.24 | 79.71 | 74.99 | 90.92 | 81.85 |
| SBERT-L-LC | **85.41** | **91.64** | **95.61** | **91.15** | **51.84** | **89.03** | **79.03** | **91.46** | **84.40** |
| SRoBERTa-L | 86.85 | 90.83 | 93.16 | 90.69 | 50.82 | 83.14 | 77.36 | 92.44 | 83.16 |
| SRoBERTa-L-LC | **87.95** | **92.49** | **95.35** | **92.06** | **54.23** | **88.51** | **79.56** | **92.94** | **85.39** |
| SupSimCSE-BL | 85.47 | **90.69** | 95.01 | 90.38 | 51.16 | 84.28 | 73.70 | 90.83 | 82.69 |
| SupSimCSE-BL-LC | **85.60** | **90.69** | **95.88** | **91.30** | **52.20** | **89.47** | **77.01** | **91.39** | **84.19** |
| SupSimCSE-RL | 88.00 | **90.69** | 94.80 | 90.86 | 51.89 | 86.52 | 74.21 | 92.84 | 83.73 |
| SupSimCSE-RL-LC | **89.24** | **90.69** | **96.29** | **92.01** | **55.39** | **90.19** | **79.82** | **93.49** | **85.89** |
| SupSimCSE-RL$_M$ | 87.78 | 92.10 | 94.72 | 90.51 | 52.43 | 83.85 | 74.39 | 92.60 | 83.55 |
| SupSimCSE-RL$_M$-LC | **89.09** | **93.44** | **96.41** | **91.43** | **56.10** | **88.06** | **78.97** | **93.44** | **85.87** |

(b) Supervised Models

Table 1: Transfer task results for different sentence embedding models (measured as accuracy * 100)

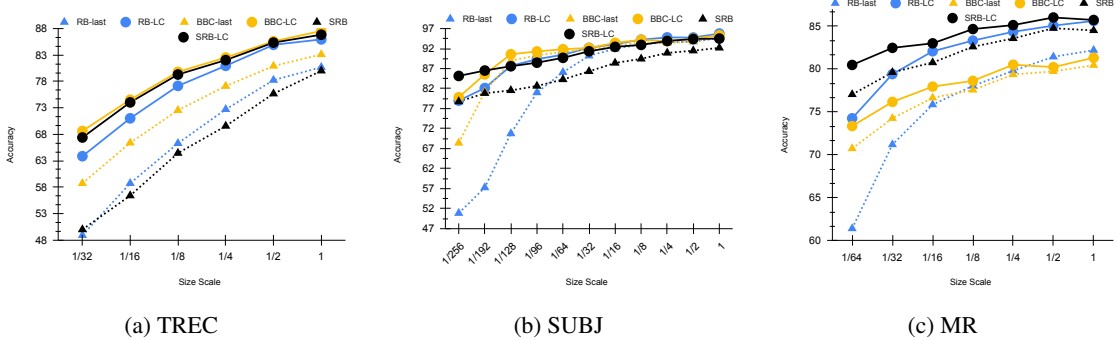

(a) TREC     (b) SUBJ     (c) MR

Figure 1: Varying the size of the training data for transfer tasks.

# 5 Ablation Studies

While more ablation studies, such as the effect of the number of layers in layer combination and the effect of excluding a particular layer from layer combination, can be found in Appendix E, in this section, we discuss the performance of layer combination in limited data settings. For this, we reduce the size of training data for MR, SUBJ, and TREC, while keeping the test data untouched. Fig. 1 shows the results for RoBERTA-B (RB), BERT-B$_{cased}$ (BBC) and SRoBERTa-B (SRB) on these data sets. We report the average results of 10 different seeds. As we can see, for all the models on all the data sets, the more reduction in size is done, the more the difference is between the LC version and the last layer average. For instance, for RB on SUBJ, on the original training data, the accuracies for RB-LC and RB are 94.77% and 93.73%, respectively, while when reducing the size to 1/256th, the respective accuracies are 78.99% and 50.83%. This suggests that it is even more beneficial to use LC when one has small training data.

# 6 Conclusion and Future Work

In this paper, we proposed a new method called BERT Layer Combination, a simple, yet effective framework, which when applied to various BERT-based models, significantly improves them for the downstream tasks of STS and transfer learning. Further, it achieves the state-of-the-art performances on eight transfer tasks. Our method combines certain layers of BERT-based models in an unsupervised manner, which shows that different layers of BERT hold important information which was previously ignored. We demonstrated the effectiveness of our approach by conducting comprehensive experiments on various BERT-based models and on a host of different tasks and data sets.

As future work, we would like to apply the technique of layer combination to other NLP tasks (e.g., punctuation insertion (Hosseini and Sameti, 2017) and question answering (Qu et al., 2019)) and also utilize it in other domains where deep learning models are used (e.g., biosignal analysis (Munia et al., 2020, Munia et al., 2023) and image captioning

([Huang et al., 2019](#))).

# 7 Limitation

One limitation of our work is that though we applied our layer combination technique to a raft of different models, all of them are BERT based; ergo, whether or not this technique works on other deep learning NLP models remains unsettled. We leave this experimentation as future work.

# 8 Acknowledgement

This research was supported in part by NSF awards DMS-1737978, DGE-2039542, OAC-1828467, OAC- 1931541, and DGE-1906630. The material presented here is partially based on High Performance Computing (HPC) resources supported by the University of Arizona TRIF, UITS, and Research, Innovation, and Impact (RII) and maintained by the UArizona Research Technologies department.

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

## A  Optimized Algorithm for Finding the Best Layer Combination

For a data set $X = \{(x_1, y_1), \ldots, (x_{|X|}, y_{|X|})\}$ where each sample consists of a list of sentences $x_i = \{S_i^1, \ldots, S_i^n\}$ ($n$=2 for sentence-pair tasks such as STS) and a label $y_i$, the best-performing layers set ($D^*$) is calculated by the following equation:

$$D^* = \underset{D \in \mathcal{P}^+(A)}{\operatorname{argmax}} m\left((f_1, \ldots, f_{|X|}), (y_1, \ldots, y_{|X|})\right)$$

(2)

where $m$ is the desired metric function, and $f_i$ is

$$f_i = f\left(p\left(H_i^1, D\right), \ldots, p\left(H_i^n, D\right)\right)$$

(3)

in which $H_j^i$ is the tensor of the $i$th sample's $j$th sentence, and $f$ is a function, such as cosine similarity.

Eq. 3 requires calculating $p(H, D)$ for each sentence in $X$ and for every possible $D$, which can be very time-consuming. In this subsection, we discuss our proposed algorithm for efficiently calculating all $p(H, D)$s in a step-by-step manner. Each method improves over the previous one, with Method 1 being the naive algorithm. The algorithms discussed here utilize *mean* as the pooling function, yet they can be easily modified to use other functions such as *max*.

**Method 1**: For every $D \in \mathcal{P}^+(A) = \mathcal{P}(A) - \emptyset$, calculate $p(H, D)$.

**Method 2**: For every $D \in \{\{i\}|0 \leq i \leq L\}$ calculate $p_l = p(H, D) = \frac{1}{N}\sum_{n=1}^{N} H_{l,n,:}$. Then, for every other $D \in \mathcal{P}^+(A)$, calculate $p(H, D)$ as $p(H, D) = \frac{1}{|D|}\sum_{l \in D} p_l$.

**Method 3**: This method is explained in Algos. 1 and 2 ($maxOptim$=False). The idea is that for calculating every $p(H, D)$, we can exploit other $P(H, D')$s that we have calculated before. For instance, $p(H, \{1, 3, 6, 8, 10\}) = \frac{3*p(H,\{1,3,6\})+2*p(H,\{8,10\})}{5}$. This method uses bottom-up dynamic programming to store the previous $p(H, D)$s. $powerset(A, i)$ in Algo. 1 is equal to $\{D|D \in \mathcal{P}(A) \wedge |D| = i\}$. $mem$ is a hash map with key-value pairs as $(D, p(H, D))$. $greedyPart(D, mem)$ returns an array of partitions of $D$ such that all partitions are present in $mem$, and each partition has the highest length possible. This is done in a greedy manner, by first

dividing $D$ into two partitions, and then three partitions, and so on. If no such partitions exist, the resultant array is empty.

**Method 4**: This method further improves over Method 3 by avoiding certain multiplications (Algo. 2, $maxOptim$=True). For instance, if $\alpha = p(H, \{1, \ldots, 4\})$ and $\beta = p(H, \{5, \ldots, 8\})$, then $p(H, \{1, \ldots, 8\})$ can be calculated as $\frac{\alpha+\beta}{2}$ instead of $\frac{\alpha*4+\beta*4}{8}$, reducing two vector multiplications. $getPartLens(parts)$ in Algo. 2 returns a *set* of the lengths of $parts$' elements.

---

**Algorithm 1:** Layer Combination Iterator

**input** : Tensor $H$, Set $A$, Bool $maxOptim$, Int $maxMem$

**output** : Set $D$, Array $P$

1 from Algo. 2 use `pool`

2 **Function** `layerIterator`($H$, $A$, $maxOptim$, $maxMem$):

3    $mem$ = `hashMap()`

4    **for** $i = 1$ *to A.length* **do**

5      **for** $D$ *in* `powerset`($A$, $i$) **do**

6        $P$ = `pool`($H$, $D$, $mem$, $maxMem$)

7        yield $D$, $P$

---

To show the effectiveness of our algorithm, we carried out an experiment. We randomly select 1000 samples from the SICK dataset and try to find the best layer combination among all possible 8192 layer combinations, once with our algorithm and once with the naive algorithm (Method 2). We do this experimentation five times and report the average results. Our algorithm takes 5.65 seconds to find the best layer combination after the tensor of all layer and token vectors are obtained by BERT-base-uncased, while Method 2 takes 1067 seconds. The one-time forward pass of BERT takes 10 seconds.

## B  STS Tasks' Experiments and Results

### B.1  Data Sets and Evaluation Setup

For STS tasks, we use the seven standard STS data sets: **STS2012-2016** (Agirre et al., 2012, 2013, 2014, 2015, 2016), **STS Benchmark** (Cer et al., 2017), and **SICK Relatedness** (Marelli et al., 2014). These data sets consist of sentence pairs and a similarity score from 1 to 5 assigned to each one of them. For each data set, we first combine all of its data subsets, shuffle it, and choose a small

**Algorithm 2:** Layer Combination Pooler

---

**input** : Tensor $H$, Set $D$, HashMap $mem$,
        Bool $maxOptim$, Int $maxMem$
**output** : Array $P$

---

1  **Function** `pool`($H$, $D$, $mem$, $maxOptim$, $maxMem$)**:**
2    $parts$ = `greedyPart`($D$, $mem$)
3    **if** $parts$ != [] **then**
4       $P$ = `array`($parts$ [0].length, 0) ;
        `// array(`$N$`,`$v$`)` `returns an array`
        `of size` $N$ `filled with` $v$`s`
5       $denom$ = $D$.length
6       **if** $maxOptim$ **then**
7          $lens$ = `getPartLens`($parts$)
8          **foreach** $len$ in $lens$ **do**
9             $cur_P$ = `array`($P$.length, 0)
10            **foreach** $part$ of size $len$ in $parts$
            **do**
11               $cur_P$ += $mem$ [$part$]
12            **if** $lens$.length > 1 **then**
13               $cur_P$ *= $len$
14            **else**
15               $denom$ = $len$
16            $P$ += $cur_P$
17       **else**
18          **foreach** $part$ in $parts$ **do**
19             $P$ += $mem$ [$part$] * $len$
20       $P$ /= $denom$
21    **else**
22       $P$ = $p(H, D)$
23    **if** $mem$.length < $maxMem$ **then**
24       $mem$ [$part$] = $P$
25    **return** $P$

---

subset (the first 350 sentence pairs) as our development data set and the rest as our test data set. We do this splitting 5 times randomly and report the average test performance. Please note that our method does not require training data. Nonetheless, we note that our method still needs a development data set in order for it to be bootstrapped. However, the required data set can be as small as 350 samples. Furthermore, as opposed to the other models, this data is not leveraged for fine-tuning, which is very time-consuming.

We use the sentence embeddings obtained by our method or the baselines to calculate the cosine similarity between two sentences and then use Spearman's correlation ($\rho$) for evaluating the models' performances as suggested by (Reimers and Gurevych, 2019).

## B.2 Baselines and Hyper-parameters

We use the same baselines that we use for the transfer tasks except for the MLM versions of the SimCSE models as those were proposed in the SimCSE paper to strengthen their models for the transfer tasks.

For the base models with 13 layers, we iterate through all the possible layer combinations for choosing the best one, yet for the large models with 25 layers, we only iterate through all the layer combinations with up to eight layers since in our experiments, we observed that combining more than eight layers hurts the performance.

## B.3 Results and Discussion

The results for STS tasks are shown in Table 2. From this table, we observe that our method significantly outperforms its corresponding variations of BERT, RoBERTa, SBERT, and SRoBERTa baselines on all the STS data sets. For instance, for BERT-L$_{uncased}$, RoBERTa-L, and SRoBERTa-B, we obtain $\rho$ improvements of up to 25.75%, 13.94%, and 3.95% and on average 16.32%, 9.52%, and 2.75%, respectively, compared to their best baselines (last avg). SimCSE-LC versions also outperform most of their baselines by a decent margin. For example, UnSupSimCSE-BL-LC improves its baseline's $\rho$ by up to 2.30% and on average 0.70%. Finally, the best SimCSE (Sup-RL)'s performances are also improved by up to 1.14%.

We can also see that our method is more effective on unsupervised models such as BERT and RoBERTa, as we spot a higher improvement compared with the supervised models such as SBERT or SupSimCSE. This suggests that the unsupervised models' layers hold more important information than the supervised models for STS tasks. However, there is still crucial information to be exploited from all the layers of the supervised models as well.

We also observe that we have higher improvements for uncased BERT versions over their baseline compared to the cased versions. Nonetheless, after applying our method to either one, the final $\rho$s are within a 1.25% range of each other. This suggests that a lot of BERT$_{uncased}$'s information

is carried in its layers, and when exploited, it can perform more closely to BERT$_{cased}$-LC.

## C Computation Setup

To conduct the experiments discussed in this paper, we used a computer with 128GB of RAM and one Intel Core i9-9980XE CPU. Our code uses PyTorch (Paszke et al., 2019), huggingface [2], and sci-kit learn (Pedregosa et al., 2011).

## D More Baselines

In this section, we compare the results of the layer combination technique with more baselines on both STS and transfer tasks. We use the same baseline models as shown in Table 2, but instead of only comparing with the last layer for BERT/RoBERTa models and original SimCSE and SBERT/SRoBERTa models (which use CLS pooler and last layer average, respectively), we also compare the results with these two new baselines: last four layers average and random layers. The results are shown in Table 3 and Table 4. The results show that our method performs better than the new baselines as well.

## E Ablation studies

### E.1 Effect of the Number of Layers

In this subsection, we show the effect of combining only $N$ layers of a model, $N$ varying from 1 to 13 for the base models. To this end, we show this effect on the data sets STS16, STSB, and SICK, and for the models BERT-B$_{cased}$, RoBERTa-B, SBERT-B, and SRoBERTa-B. From Figs. 2 and 3, we can see that (1) For all the data sets and all the models adding more layers shows an upward trend in performance up to some peak point (e.g., $N = 4$ for BERT-B$_{cased}$ on STS16), and adding more layers after that point shows a downward trend in performance; however, this point is different for different models and data sets, yet it is always at most 6. (2) For BERT-B$_{cased}$, RoBERTa-B, and SBERT-B, moving from one layer to two layers leads to a huge increase in performances for all but one data set-model pairs, yet combining more than two layers always leads to a further substantial increase in performance. For instance, on STSB, an absolute $\rho$ improvement of 1.22% can be obtained when moving from two layers to six layers. We, nonetheless,

do not see drastic changes for SRoBERTa-B when moving from one layer to two layers.

### E.2 Effect of Excluding a Particular Layer from Layer Combination

In this section, we discuss how excluding one particular layer from layer combination affects the performances. For this, we show the results for the models BERT-B$_{cased}$, RoBERTa-B, and SBERT-B on the data sets STS16, STSB, and SICK in Fig. 4. We can mention these observations:

1. Most of the time, excluding layer 11 hurts the performance, showing that this is an important layer.

2. Excluding either of the layers 4, 5, or 6 does not hurt the performance, suggesting that these layers are not important for these models and data sets.

3. Last layer (L12) is always important for SBERT-B, is important for BERT-B$_{cased}$ in two cases (STS16 and STSB), and is only important for RoBERTa-B in one case (STSB).

4. Layer 0, which is the embedding layer, proves to be important in more than half of the cases here, showing that it carries important information to be considered for STS tasks, and it cannot be ignored.

5. Excluding any particular layer in any of the nine model-data set pairs shown in Fig. 4, leads to a maximum decrease of 0.97% (L12 for SBERT-B on STS16) in the performance. However as shown in Table 2, in any of these 9 cases, we get at least an improvement of 1.86% by combining certain layers (considering all the layers). This suggests that layer combination still outperforms the baselines even if one (any) layer is not considered at all.

---

[2] https://huggingface.co/transformers/

**Table 2 (a) Unsupervised Models**

| Model | STS12 | STS13 | STS14 | STS15 | STS16 | STSB | SICK | Avg. |
|---|---|---|---|---|---|---|---|---|
| BERT-B$_{uncased}$-CLS | 7.23 | 28.44 | 12.48 | 15.79 | 28.21 | 5.36 | 29.83 | 18.19 |
| BERT-B$_{uncased}$-last | 31.08 | 59.58 | 47.39 | 60.17 | 63.04 | 46.24 | 57.90 | 52.20 |
| BERT-B$_{uncased}-LC$ | 50.89 | 64.67 | 54.94 | 72.22 | 67.95 | 59.42 | 63.54 | 61.95 |
| BERT-B$_{cased}$-CLS | 14.16 | 22.51 | 16.41 | 24.11 | 27.52 | 12.69 | 39.22 | 22.37 |
| BERT-B$_{cased}$-last | 38.30 | 62.12 | 53.25 | 64.43 | 63.86 | 55.86 | 58.92 | 56.68 |
| BERT-B$_{cased}-LC$ | 50.55 | 66.25 | 57.91 | 72.51 | 67.90 | 62.98 | 61.57 | 62.81 |
| RoBERTa-B-last | 32.18 | 56.27 | 45.05 | 61.11 | 60.81 | 55.13 | 62.10 | 53.24 |
| RoBERTa-B-LC | 45.58 | 60.83 | 51.20 | 69.23 | 64.68 | 60.17 | 64.00 | 59.38 |
| BERT-L$_{uncased}$-CLS | 18.66 | 21.47 | 13.79 | 11.01 | 23.29 | 13.31 | 25.11 | 18.09 |
| BERT-L$_{uncased}$-last | 27.76 | 55.11 | 44.37 | 51.59 | 61.10 | 46.52 | 53.56 | 48.57 |
| BERT-L$_{uncased}-LC$ | 53.51 | 68.19 | 58.45 | 74.43 | 71.03 | 63.97 | 64.64 | 64.89 |
| BERT-L$_{cased}$-CLS | 14.64 | 8.27 | 6.04 | 9.74 | 25.00 | 10.69 | 26.99 | 14.48 |
| BERT-L$_{cased}$-last | 45.77 | 63.81 | 54.41 | 68.87 | 64.53 | 58.25 | 63.24 | 59.84 |
| BERT-L$_{cased}-LC$ | 53.87 | 70.23 | 61.04 | 75.66 | 70.70 | 66.16 | 65.22 | 66.13 |
| RoBERTa-L-last | 33.49 | 57.64 | 45.49 | 62.74 | 61.40 | 51.59 | 57.86 | 52.89 |
| RoBERTa-L-LC | 47.43 | 65.33 | 55.14 | 72.44 | 69.04 | 62.60 | 64.88 | 62.41 |
| UnSupSimCSE-BL | 69.21 | 83.93 | 75.58 | 83.86 | 78.98 | 77.89 | 73.47 | 77.56 |
| UnSupSimCSE-BL-LC | 69.21 | 84.82 | 75.88 | 84.41 | 79.85 | 80.19 | 73.47 | 78.26 |
| UnSupSimCSE-RL | 72.11 | 83.41 | 74.96 | 84.03 | 80.79 | 81.74 | 70.82 | 78.27 |
| UnSupSimCSE-RL-LC | 72.11 | 83.66 | 75.03 | 84.37 | 80.68 | 82.06 | 72.53 | 78.63 |

(a) Unsupervised Models. CLS: CLS pooler, last: last layer's average.

**Table 2 (b) Supervised Models**

| Model | STS12 | STS13 | STS14 | STS15 | STS16 | STSB | SICK | Avg. |
|---|---|---|---|---|---|---|---|---|
| SBERT-B | 70.78 | 76.69 | 73.13 | 79.08 | 74.20 | 76.66 | 72.75 | 74.76 |
| SBERT-B-LC | 72.02 | 79.18 | 74.50 | 82.63 | 76.42 | 78.52 | 76.51 | 77.11 |
| SRoBERTa-B | 70.82 | 73.06 | 70.61 | 78.34 | 74.17 | 76.65 | 74.31 | 73.99 |
| SRoBERTa-B-LC | 72.94 | 76.14 | 72.83 | 82.29 | 77.13 | 78.99 | 76.90 | 76.75 |
| SBERT-L | 72.15 | 78.49 | 74.91 | 80.93 | 76.72 | 78.82 | 73.63 | 76.52 |
| SBERT-L-LC | 72.79 | 81.39 | 76.75 | 84.14 | 79.15 | 80.43 | 77.22 | 78.84 |
| SRoBERTa-L | 74.06 | 77.04 | 73.07 | 81.59 | 76.69 | 78.24 | 74.08 | 76.40 |
| SRoBERTa-L-LC | 74.96 | 80.25 | 75.54 | 84.71 | 79.52 | 80.78 | 77.72 | 79.07 |
| SupSimCSE-BL | 75.51 | 86.56 | 80.22 | 86.09 | 81.64 | 84.86 | 80.93 | 82.26 |
| SupSimCSE-BL-LC | 75.51 | 86.56 | 80.22 | 86.09 | 82.42 | 84.86 | 80.93 | 82.37 |
| SupSimCSE-RL | 77.35 | 87.36 | 82.18 | 86.59 | 83.92 | 86.58 | 81.72 | 83.67 |
| SupSimCSE-RL-LC | 77.35 | 87.82 | 82.18 | 87.12 | 85.05 | 86.60 | 81.72 | 83.98 |

(b) Supervised Models

Table 2: Spearman's rank correlation $\rho * 100$ between the cosine similarity of sentence representations and the gold labels for STS tasks. The highest numbers among models with the same encoder are boldfaced.

**Table 3 (a) Unsupervised Models**

| Model | STS12 | STS13 | STS14 | STS15 | STS16 | STSB | SICK | Avg. |
|---|---|---|---|---|---|---|---|---|
| BERT-B$_{uncased}$-last | 31.08 | 59.58 | 47.39 | 60.17 | 63.04 | 46.24 | 57.90 | 52.20 |
| BERT-B$_{uncased}$-last 4 | 36.44 | 59.08 | 49.24 | 64.83 | 62.01 | 47.02 | 58.36 | 53.85 |
| BERT-B$_{uncased}$-rand | 45.23 | 61.48 | 51.89 | 69.54 | 61.76 | 56.73 | 62.17 | 58.40 |
| BERT-B$_{uncased}$-LC | 50.89 | 64.67 | 54.94 | 72.22 | 67.95 | 59.42 | 63.54 | 61.95 |
| BERT-B$_{cased}$-last | 38.30 | 62.12 | 53.25 | 64.43 | 63.86 | 55.86 | 58.92 | 56.68 |
| BERT-B$_{cased}$-last 4 | 38.20 | 60.32 | 51.75 | 65.95 | 62.29 | 54.11 | 59.37 | 56.00 |
| BERT-B$_{cased}$-rand | 45.52 | 62.42 | 55.90 | 71.27 | 65.62 | 57.89 | 60.34 | 59.85 |
| BERT-B$_{cased}$-LC | 50.55 | 66.25 | 57.91 | 72.51 | 67.90 | 62.98 | 61.57 | 62.81 |
| RoBERTa-B-last | 32.18 | 56.27 | 45.05 | 61.11 | 60.81 | 55.13 | 62.10 | 53.24 |
| RoBERTa-B-last 4 | 35.63 | 56.30 | 46.31 | 64.03 | 63.33 | 55.35 | 62.09 | 54.72 |
| RoBERTa-B-rand | 43.75 | 58.88 | 47.54 | 68.56 | 63.81 | 58.69 | 62.51 | 57.68 |
| RoBERTa-B-LC | 45.58 | 60.83 | 51.20 | 69.23 | 64.68 | 60.17 | 64.00 | 59.38 |
| BERT-L$_{uncased}$-last | 27.76 | 55.11 | 44.37 | 51.59 | 61.10 | 46.52 | 53.56 | 48.57 |
| BERT-L$_{uncased}$-last 4 | 34.91 | 58.09 | 49.01 | 59.16 | 61.30 | 48.95 | 54.56 | 52.28 |
| BERT-L$_{uncased}$-rand | 45.09 | 62.32 | 53.19 | 71.18 | 67.35 | 60.51 | 62.02 | 60.24 |
| BERT-L$_{uncased}$-LC | 53.51 | 68.19 | 58.45 | 74.43 | 71.03 | 63.97 | 64.64 | 64.89 |
| BERT-L$_{cased}$-last | 45.77 | 63.81 | 54.41 | 68.87 | 64.53 | 58.25 | 63.24 | 59.84 |
| BERT-L$_{cased}$-last 4 | 43.52 | 61.48 | 52.87 | 68.32 | 63.78 | 54.70 | 63.15 | 58.26 |
| BERT-L$_{cased}$-rand | 52.30 | 65.77 | 52.46 | 72.25 | 68.80 | 62.70 | 62.92 | 62.46 |
| BERT-L$_{cased}$-LC | 53.87 | 70.23 | 61.04 | 75.66 | 70.70 | 66.16 | 65.22 | 66.13 |
| RoBERTa-L-last | 33.49 | 57.64 | 45.49 | 62.74 | 61.40 | 51.59 | 57.86 | 52.89 |
| RoBERTa-L-last 4 | 34.28 | 59.01 | 47.27 | 64.86 | 66.23 | 57.98 | 62.24 | 55.98 |
| RoBERTa-L-rand | 44.52 | 61.20 | 52.47 | 70.37 | 68.14 | 61.47 | 63.85 | 60.29 |
| RoBERTa-L-LC | 47.43 | 65.33 | 55.14 | 72.44 | 69.04 | 62.60 | 64.88 | 62.41 |
| UnSupSimCSE-BL | 69.21 | 83.93 | 75.58 | 83.86 | 78.98 | 77.89 | 73.47 | 77.56 |
| UnSupSimCSE-BL-last 4 | 64.83 | 83.43 | 74.94 | 83.79 | 78.16 | 78.14 | 72.61 | 76.56 |
| UnSupSimCSE-BL-rand | 52.67 | 64.63 | 56.26 | 75.56 | 66.91 | 69.87 | 66.78 | 64.67 |
| UnSupSimCSE-BL-LC | 69.21 | 84.82 | 75.88 | 84.41 | 79.85 | 80.19 | 73.47 | 78.26 |
| UnSupSimCSE-RL | 72.11 | 83.41 | 74.96 | 84.03 | 80.79 | 81.74 | 70.82 | 78.27 |
| UnSupSimCSE-RL-last 4 | 69.44 | 83.28 | 74.89 | 83.95 | 80.21 | 82.00 | 71.55 | 77.90 |
| UnSupSimCSE-RL-rand | 59.92 | 79.44 | 67.06 | 81.83 | 75.26 | 59.52 | 67.31 | 70.05 |
| UnSupSimCSE-RL-LC | 72.11 | 83.66 | 75.03 | 84.37 | 80.68 | 82.06 | 72.53 | 78.63 |

(a) Unsupervised Models. last: last layer's average.

**Table 3 (b) Supervised Models**

| Model | STS12 | STS13 | STS14 | STS15 | STS16 | STSB | SICK | Avg. |
|---|---|---|---|---|---|---|---|---|
| SBERT-B | 70.78 | 76.69 | 73.13 | 79.08 | 74.20 | 76.66 | 72.75 | 74.76 |
| SBERT-B-last 4 | 69.86 | 77.50 | 73.14 | 78.97 | 74.27 | 77.25 | 72.38 | 74.77 |
| SBERT-B-rand | 66.91 | 72.83 | 66.02 | 70.65 | 72.92 | 70.47 | 67.60 | 69.63 |
| SBERT-B-LC | 72.02 | 79.18 | 74.50 | 82.63 | 76.42 | 78.52 | 76.51 | 77.11 |
| SRoBERTa-B | 70.82 | 73.06 | 70.61 | 78.34 | 74.17 | 76.65 | 74.31 | 73.99 |
| SRoBERTa-B-last 4 | 71.31 | 74.64 | 71.76 | 77.61 | 73.88 | 76.92 | 74.16 | 74.33 |
| SRoBERTa-B-rand | 59.44 | 74.73 | 71.22 | 78.89 | 65.09 | 67.33 | 75.33 | 70.29 |
| SRoBERTa-B-LC | 72.94 | 76.14 | 72.83 | 82.29 | 77.13 | 78.99 | 76.90 | 76.75 |
| SBERT-L | 72.15 | 78.49 | 74.91 | 80.93 | 76.72 | 78.82 | 73.63 | 76.52 |
| SBERT-L-last 4 | 70.03 | 78.40 | 74.82 | 80.05 | 73.97 | 77.98 | 72.54 | 75.40 |
| SBERT-L-rand | 63.69 | 74.69 | 61.10 | 83.55 | 71.49 | 76.06 | 73.70 | 72.04 |
| SBERT-L-LC | 72.79 | 81.39 | 76.75 | 84.14 | 79.15 | 80.43 | 77.22 | 78.84 |
| SRoBERTa-L | 74.06 | 77.04 | 73.07 | 81.59 | 76.69 | 78.24 | 74.08 | 76.40 |
| SRoBERTa-L-last 4 | 71.52 | 77.67 | 73.35 | 79.28 | 75.73 | 77.98 | 72.85 | 75.48 |
| SRoBERTa-L-rand | 54.05 | 62.17 | 66.58 | 75.98 | 68.35 | 79.03 | 76.83 | 69.00 |
| SRoBERTa-L-LC | 74.96 | 80.25 | 75.54 | 84.71 | 79.52 | 80.78 | 77.72 | 79.07 |
| SupSimCSE-BL | 75.51 | 86.56 | 80.22 | 86.09 | 81.64 | 84.86 | 80.93 | 82.26 |
| SupSimCSE-BL-last 4 | 71.16 | 85.17 | 77.17 | 85.19 | 80.80 | 83.14 | 80.18 | 80.40 |
| SupSimCSE-BL-rand | 52.84 | 71.87 | 65.08 | 76.55 | 69.46 | 64.42 | 68.65 | 66.98 |
| SupSimCSE-BL-LC | 75.51 | 86.56 | 80.22 | 86.09 | 82.42 | 84.86 | 80.93 | 82.37 |
| SupSimCSE-RL | 77.35 | 87.36 | 82.18 | 86.59 | 83.92 | 86.58 | 81.72 | 83.67 |
| SupSimCSE-RL-last 4 | 75.64 | 87.58 | 81.41 | 85.92 | 83.73 | 86.20 | 80.54 | 83.00 |
| SupSimCSE-RL-rand | 70.06 | 85.95 | 56.54 | 83.06 | 82.04 | 80.93 | 69.46 | 75.43 |
| SupSimCSE-RL-LC | 77.35 | 87.82 | 82.18 | 87.12 | 85.05 | 86.60 | 81.72 | 83.98 |

(b) Supervised Models

Table 3: STS Task's Results with more baselines. The same metric as in Table 2 is used ($\rho * 100$). last 4 is the average of the last 4 layers, and rand is the random layers combination.

**(a) Unsupervised Models. last: last layer's average.**

| Model | MR | CR | SUBJ | MPQA | SSTM | TREC | MRPC | SST | Avg. |
|---|---|---|---|---|---|---|---|---|---|
| BERT-B$_{uncased}$-last | 81.19 | 86.77 | 95.08 | 87.97 | 47.48 | 85.76 | 73.66 | 87.20 | 80.64 |
| BERT-B$_{uncased}$-last 4 | 82.05 | 86.74 | 95.45 | 88.85 | 48.38 | 87.61 | 75.72 | 87.88 | 81.59 |
| BERT-B$_{uncased}$-rand | 80.89 | 84.69 | 94.46 | 88.8 | 47.09 | 85.39 | 76.49 | 85.74 | 80.44 |
| BERT-B$_{uncased}$-LC | 82.24 | 87.20 | 95.45 | 90.12 | 49.01 | 88.76 | 77.08 | 88.22 | 82.26 |
| BERT-B$_{cased}$-last | 80.55 | 85.19 | 94.64 | 87.82 | 45.69 | 83.65 | 74.30 | 85.83 | 79.71 |
| BERT-B$_{cased}$-last 4 | 81.38 | 86.17 | 94.89 | 88.55 | 46.75 | 87.10 | 73.95 | 86.45 | 80.66 |
| BERT-B$_{cased}$-rand | 79.41 | 84.12 | 94.06 | 89.10 | 45.38 | 86.11 | 73.57 | 85.65 | 79.68 |
| BERT-B$_{cased}$-LC | 81.42 | 86.98 | 95.11 | 90.08 | 46.99 | 88.00 | 75.45 | 86.88 | 81.36 |
| RoBERTa-B-last | 82.58 | 84.69 | 94.55 | 85.28 | 50.10 | 81.90 | 72.51 | 87.28 | 79.86 |
| RoBERTa-B-last 4 | 85.01 | 88.04 | 95.44 | 87.53 | 51.30 | 85.85 | 71.93 | 88.83 | 81.74 |
| RoBERTa-B-rand | 84.41 | 87.15 | 94.78 | 87.56 | 50.08 | 85.08 | 72.12 | 88.31 | 81.19 |
| RoBERTa-B-LC | 85.89 | 90.23 | 95.65 | 88.66 | 52.48 | 86.92 | 74.85 | 89.65 | 83.04 |
| UnSupSimCSE-BB | 80.67 | 85.29 | 94.29 | 88.75 | 46.13 | 83.96 | 72.55 | 85.86 | 79.69 |
| UnSupSimCSE-BB-last 4 | 81.64 | 85.93 | 95.01 | 89.52 | 47.95 | 87.73 | 76.97 | 87.38 | 81.52 |
| UnSupSimCSE-BB-rand | 76.46 | 84.47 | 94.05 | 89.52 | 46.78 | 86.62 | 73.75 | 86.52 | 79.77 |
| UnSupSimCSE-BB-LC | 81.61 | 86.77 | 95.23 | 90.12 | 48.25 | 89.05 | 77.36 | 87.55 | 81.99 |
| UnSupSimCSE-RB | 80.85 | 85.36 | 92.40 | 86.70 | 47.25 | 76.13 | 72.90 | 85.67 | 78.41 |
| UnSupSimCSE-RB-last 4 | 83.09 | 87.8 | 94.49 | 88.15 | 49.48 | 84.01 | 77.86 | 87.27 | 81.52 |
| UnSupSimCSE-RB-rand | 82.05 | 86.70 | 93.98 | 88.26 | 49.64 | 83.73 | 75.94 | 87.28 | 80.95 |
| UnSupSimCSE-RB-LC | 84.00 | 88.99 | 94.72 | 88.67 | 50.52 | 86.02 | 78.07 | 88.33 | 82.41 |
| BERT-L$_{uncased}$-last | 83.88 | 88.29 | 95.52 | 86.51 | 49.89 | 83.81 | 71.49 | 88.48 | 80.98 |
| BERT-L$_{uncased}$-last 4 | 84.90 | 89.91 | 95.95 | 87.23 | 49.83 | 86.72 | 73.15 | 89.16 | 82.11 |
| BERT-L$_{uncased}$-rand | 83.29 | 88.04 | 95.05 | 89.38 | 49.54 | 86.62 | 75.68 | 89.03 | 82.08 |
| BERT-L$_{uncased}$-LC | 85.21 | 89.88 | 96.08 | 90.17 | 51.06 | 88.89 | 76.57 | 89.96 | 83.48 |
| BERT-L$_{cased}$-last | 84.28 | 88.68 | 94.95 | 88.03 | 49.23 | 84.10 | 72.62 | 88.75 | 81.33 |
| BERT-L$_{cased}$-last 4 | 85.14 | 89.17 | 95.28 | 89.13 | 50.59 | 86.52 | 74.67 | 89.43 | 82.49 |
| BERT-L$_{cased}$-rand | 82.62 | 87.58 | 94.58 | 89.86 | 49.27 | 86.02 | 76.10 | 88.79 | 81.85 |
| BERT-L$_{cased}$-LC | 85.34 | 90.19 | 95.41 | 90.50 | 50.92 | 88.62 | 77.17 | 90.00 | 83.52 |
| RoBERTa-L-last | 84.30 | 85.22 | 94.93 | 87.25 | 50.59 | 81.75 | 67.24 | 89.41 | 80.09 |
| RoBERTa-L-last 4 | 86.03 | 88.61 | 95.85 | 89.52 | 51.91 | 86.52 | 69.49 | 90.74 | 82.33 |
| RoBERTa-L-rand | 82.85 | 89.34 | 95.51 | 90.27 | 52.88 | 86.78 | 70.33 | 90.95 | 82.36 |
| RoBERTa-L-LC | 88.04 | 91.68 | 96.51 | 91.11 | 54.11 | 88.06 | 75.01 | 92.08 | 84.58 |
| UnSupSimCSE-BL | 84.85 | 88.15 | 95.09 | 89.13 | 48.84 | 83.63 | 74.09 | 89.02 | 81.60 |
| UnSupSimCSE-BL-last 4 | 83.91 | 89.38 | 95.60 | 90.14 | 48.87 | 86.79 | 75.49 | 89.70 | 82.48 |
| UnSupSimCSE-BL-rand | 82.90 | 86.87 | 94.90 | 90.30 | 49.28 | 86.31 | 75.78 | 88.51 | 81.86 |
| UnSupSimCSE-BL-LC | 84.85 | 89.84 | 95.77 | 90.62 | 50.93 | 89.07 | 77.10 | 90.04 | 83.53 |
| UnSupSimCSE-RL | 82.29 | 86.24 | 92.77 | 88.12 | 45.98 | 82.02 | 73.91 | 88.08 | 79.93 |
| UnSupSimCSE-RL-last 4 | 84.81 | 89.45 | 94.61 | 88.94 | 48.97 | 86.47 | 78.97 | 89.98 | 82.78 |
| UnSupSimCSE-RL-rand | 84.64 | 89.48 | 91.58 | 89.91 | 51.46 | 87.00 | 74.19 | 90.29 | 82.32 |
| UnSupSimCSE-RL-LC | 86.75 | 91.01 | 95.72 | 90.85 | 52.83 | 87.93 | 79.13 | 91.57 | 84.47 |

**(b) Supervised Models**

| Model | MR | CR | SUBJ | MPQA | SSTM | TREC | MRPC | SST | Avg. |
|---|---|---|---|---|---|---|---|---|---|
| SBERT-B | 82.90 | 89.21 | 93.93 | 89.79 | 48.14 | 80.04 | 74.30 | 89.19 | 80.94 |
| SBERT-B-last 4 | 83.56 | 89.74 | 95.04 | 90.17 | 49.49 | 85.17 | 77.91 | 89.54 | 82.58 |
| SBERT-B-rand | 82.70 | 89.20 | 94.47 | 89.52 | 49.12 | 85.91 | 77.61 | 88.92 | 82.18 |
| SBERT-B-LC | 83.61 | 90.05 | 95.48 | 91.02 | 49.48 | 88.56 | 78.60 | 89.68 | 83.27 |
| SRoBERTa-B | 84.67 | 90.12 | 92.57 | 89.3 | 50.64 | 81.79 | 77.43 | 90.12 | 82.08 |
| SRoBERTa-B-last 4 | 85.81 | 90.97 | 93.53 | 89.87 | 53.13 | 85.31 | 78.64 | 90.54 | 83.47 |
| SRoBERTa-B-rand | 85.05 | 90.47 | 94.09 | 89.91 | 52.33 | 84.61 | 74.56 | 90.15 | 82.65 |
| SRoBERTa-B-LC | 85.76 | 91.71 | 94.89 | 90.62 | 53.69 | 87.95 | 78.92 | 90.78 | 84.29 |
| SupSimCSE-BB | 81.85 | 89.56 | 94.60 | 89.92 | 50.16 | 82.75 | 74.60 | 88.81 | 81.53 |
| SupSimCSE-BB-last 4 | 82.67 | 89.14 | 95.48 | 90.74 | 49.78 | 87.61 | 77.33 | 89.50 | 82.78 |
| SupSimCSE-BB-rand | 76.70 | 87.26 | 94.31 | 90.29 | 48.91 | 88.08 | 76.95 | 88.09 | 81.32 |
| SupSimCSE-BB-LC | 81.85 | 89.56 | 95.68 | 90.77 | 50.16 | 89.09 | 77.52 | 89.71 | 83.04 |
| SupSimCSE-RB | 84.29 | 91.11 | 93.12 | 90.19 | 52.69 | 81.12 | 76.14 | 90.18 | 82.36 |
| SupSimCSE-RB-last 4 | 85.64 | 92.10 | 95.00 | 90.76 | 53.58 | 87.35 | 76.60 | 90.86 | 83.99 |
| SupSimCSE-RB-rand | 84.44 | 90.61 | 94.07 | 90.40 | 53.01 | 85.14 | 72.49 | 90.32 | 82.56 |
| SupSimCSE-RB-LC | 86.00 | 92.13 | 95.21 | 90.87 | 53.48 | 87.97 | 78.76 | 91.02 | 84.43 |
| SupSimCSE-RB$_M$ | 84.56 | 91.89 | 93.29 | 89.52 | 52.47 | 81.88 | 76.23 | 90.13 | 82.50 |
| SupSimCSE-RB$_M$-last 4 | 85.79 | 92.56 | 95.60 | 90.43 | 53.65 | 87.82 | 74.39 | 90.91 | 83.89 |
| SupSimCSE-RB$_M$-rand | 84.91 | 90.72 | 94.49 | 89.88 | 53.23 | 87.90 | 72.79 | 89.87 | 82.97 |
| SupSimCSE-RB$_M$-LC | 86.11 | 92.45 | 95.81 | 90.62 | 54.15 | 88.40 | 78.14 | 91.00 | 84.59 |
| SBERT-L | 84.69 | 90.62 | 94.40 | 90.25 | 49.24 | 79.71 | 74.99 | 90.92 | 81.85 |
| SBERT-L-last 4 | 85.12 | 91.04 | 95.20 | 90.46 | 51.06 | 84.52 | 78.00 | 91.33 | 83.34 |
| SBERT-L-rand | 84.31 | 91.14 | 94.50 | 90.59 | 49.92 | 85.12 | 78.53 | 86.19 | 82.54 |
| SBERT-L-LC | 85.41 | 91.64 | 95.61 | 91.15 | 51.84 | 89.03 | 79.03 | 91.46 | 84.40 |
| SRoBERTa-L | 86.85 | 90.83 | 93.16 | 90.69 | 50.82 | 83.14 | 77.36 | 92.44 | 83.16 |
| SRoBERTa-L-last 4 | 87.80 | 91.50 | 94.21 | 91.06 | 52.02 | 84.66 | 77.89 | 92.58 | 83.97 |
| SRoBERTa-L-rand | 85.81 | 84.19 | 94.23 | 91.34 | 52.50 | 86.44 | 77.55 | 92.12 | 83.02 |
| SRoBERTa-L-LC | 87.95 | 92.49 | 95.35 | 92.06 | 54.23 | 88.51 | 79.56 | 92.94 | 85.39 |
| SupSimCSE-BL | 85.47 | 90.69 | 95.01 | 90.38 | 51.16 | 84.28 | 73.70 | 90.83 | 82.69 |
| SupSimCSE-BL-last 4 | 84.84 | 90.26 | 95.56 | 90.76 | 51.38 | 86.63 | 75.68 | 91.08 | 83.27 |
| SupSimCSE-BL-rand | 83.64 | 89.66 | 94.69 | 90.73 | 50.88 | 86.94 | 74.70 | 90.52 | 82.72 |
| SupSimCSE-BL-LC | 85.60 | 90.69 | 95.88 | 91.30 | 52.20 | 89.47 | 77.01 | 91.39 | 84.19 |
| SupSimCSE-RL | 88.00 | 90.97 | 94.80 | 90.86 | 51.89 | 86.52 | 74.21 | 92.84 | 83.76 |
| SupSimCSE-RL-last 4 | 88.84 | 91.99 | 95.85 | 91.36 | 53.15 | 88.35 | 79.72 | 93.39 | 85.33 |
| SupSimCSE-RL-rand | 86.39 | 91.67 | 95.45 | 91.47 | 54.56 | 88.48 | 72.65 | 92.97 | 84.21 |
| SupSimCSE-RL-LC | 89.24 | 92.20 | 96.29 | 92.01 | 55.39 | 90.19 | 79.82 | 93.49 | 86.08 |
| SupSimCSE-RL$_M$ | 87.78 | 92.10 | 94.72 | 90.51 | 52.43 | 83.85 | 74.39 | 92.60 | 83.55 |
| SupSimCSE-RL$_M$-last 4 | 88.72 | 93.16 | 95.91 | 91.11 | 54.20 | 86.79 | 77.86 | 93.29 | 85.13 |
| SupSimCSE-RL$_M$-rand | 87.74 | 92.73 | 95.53 | 88.76 | 55.46 | 86.96 | 72.44 | 92.79 | 84.05 |
| SupSimCSE-RL$_M$-LC | 89.09 | 93.44 | 96.41 | 91.43 | 56.10 | 88.06 | 78.97 | 93.44 | 85.87 |

Table 4: Transfer Task's Results with more baselines. The same metric as in Table 1 is used (accuracy). last 4 is the average of the last 4 layers, and rand is the random layers combination.

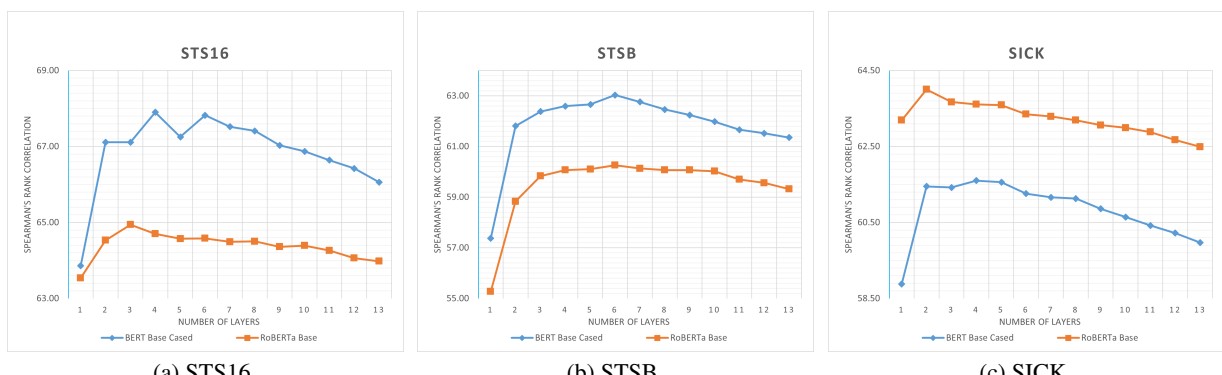

(a) STS16     (b) STSB     (c) SICK

Figure 2: Effect of combining $N$ Layers for BERT-B$_{cased}$ and RoBERTa-B

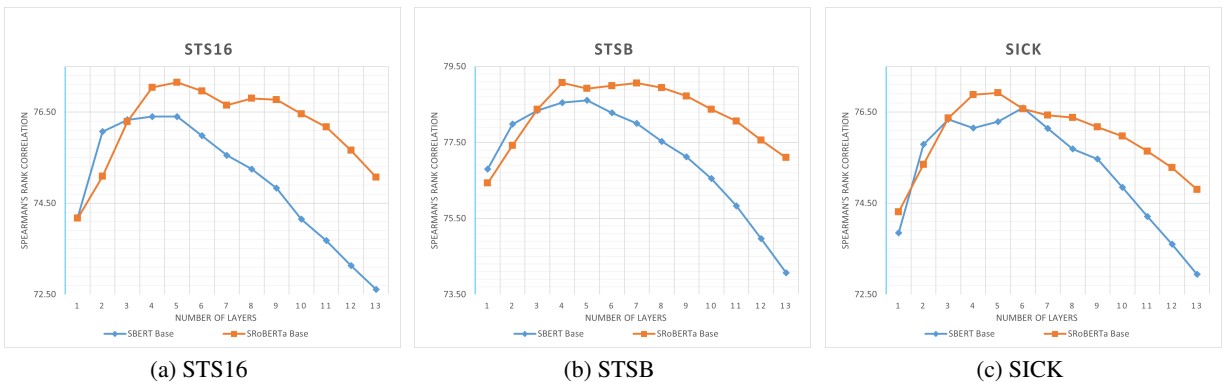

(a) STS16            (b) STSB            (c) SICK

Figure 3: Effect of combining $N$ Layers for SBERT-B and SRoBERTa

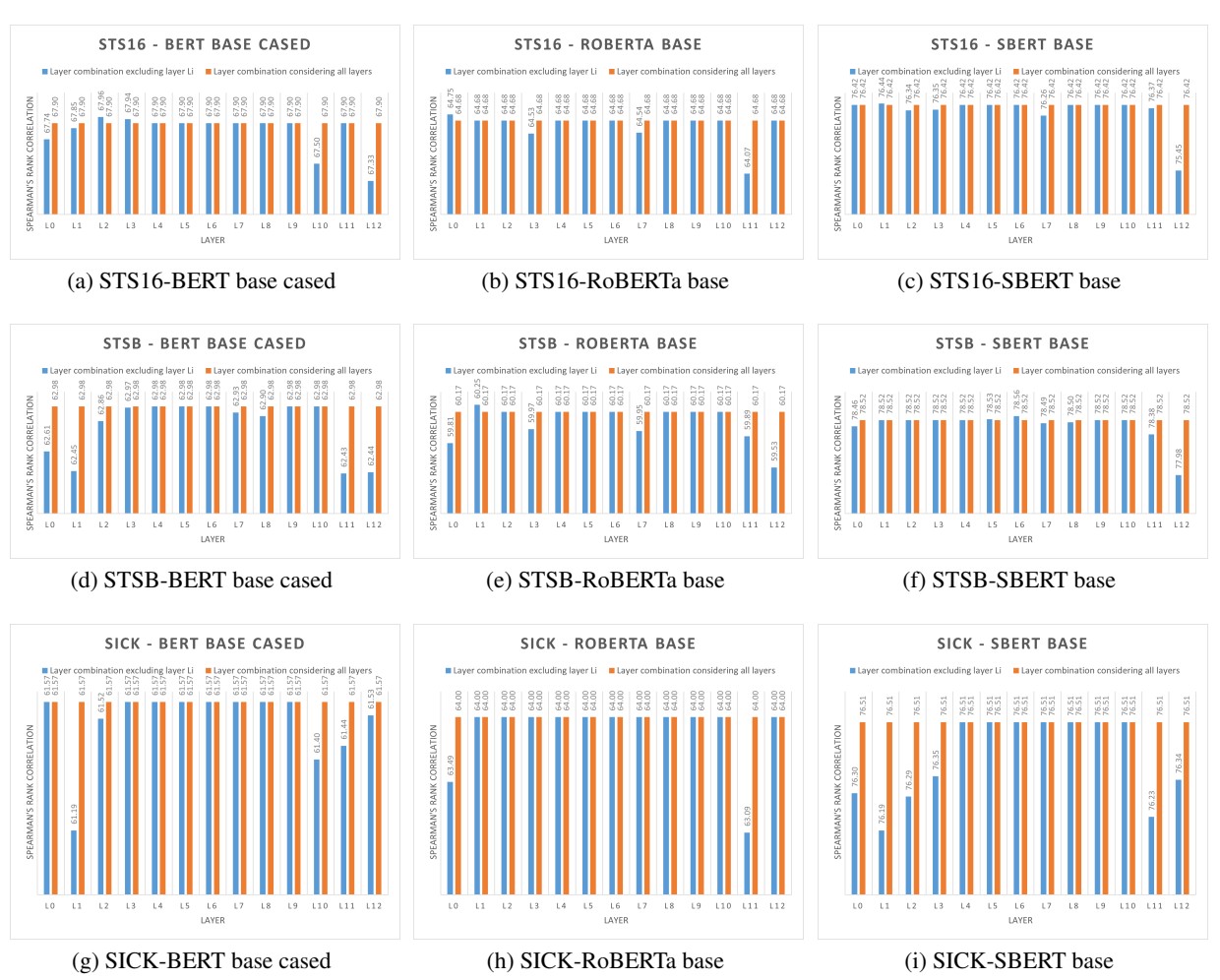

(a) STS16-BERT base cased      (b) STS16-RoBERTa base      (c) STS16-SBERT base

(d) STSB-BERT base cased      (e) STSB-RoBERTa base      (f) STSB-SBERT base

(g) SICK-BERT base cased      (h) SICK-RoBERTa base      (i) SICK-SBERT base

Figure 4: Effect of Excluding a Particular Layer