# OpenReview forum: "BERT Has More to Offer: BERT Layers Combination Yields Better Sentence Embeddings"
_EMNLP/2023/Conference — EMNLP 2023 Findings_

### Official Review · Reviewer_ciSn · 2023-08-02

**Soundness:** 3

**Excitement:**

3: Ambivalent: It has merits (e.g., it reports state-of-the-art results, the idea is nice), but there are key weaknesses (e.g., it describes incremental work), and it can significantly benefit from another round of revision. However, I won't object to accepting it if my co-reviewers champion it.

**Paper Topic And Main Contributions:**

This paper proposes a new method called BERT-LC (BERT Layers Combination) that obtains sentence embeddings by combining certain layers of BERT-based models in an unsupervised way. The authors argue that different layers of BERT carry different features, such as surface, syntactic, and semantic, and that each data set and task might need a different set of features for its sentences. The authors show that their method outperforms the baseline BERT and other models on seven semantic textual similarity (STS) data sets and eight transfer tasks. They also develop an algorithm that speeds up the process of finding the best layer combination and propose an innovative method that integrates the layer combination method with the CLS pooling head.

**Questions For The Authors:**

1.	In line 147, do you experiment on different pooling function? Can you report the experiments results over different pooling function?

2.	In Appendix A, can you provide the complete time comparison of different methods? I would appreciate a thorough analysis of effectiveness of different improvement.

3.	Can you provide the details about two-step pipeline mentioned in Line 182?


**Reasons To Accept:**

It is interesting to see some theoretical or empirical explanation for why combining certain layers of BERT works better than others.

1.	The proposed method is concise and effective compared to the selected baselines.

2.	The author conduct a substantial number of experiments on multiple pre-trained language models to show the effectiveness of the proposed method.


**Reasons To Reject:**

1.	The writing is difficult to follow. For instance, lines 001-006 in the abstract are not clear enough. The author seems to suggest that the proposed method does not require training, but this point is not emphasized throughout the entire paper until line 211. Lines 103-106 are also confusing and hard to understand. The author should carefully examine each sentence in the paper to ensure they are easily understandable.

2.	The use of mathematical symbols in this paper is also challenging to read, such as the writing in lines 217-224, which is not very clear. Many mathematical symbols are not explained, for example, in line 163, it is not clear what $\mathcal{P}(A)$ and $\emptyset$ refer to (though can be gussed). Similarly, lines 664-686 also have the same issue, and they are difficult to comprehend.

3.	The authors should carefully format Algorithm 1 and Algorithm 2 to make them more easily readable.

4.	The paper lacks a comparison with fine-tuned models, for instance, on the SST dataset, where fine-tuning bert-base can easily achieve over 93% accuracy. However, the results in this paper only reach 88%, which is not the state-of-the-art as claimed by the authors.


**Reproducibility:**

3: Could reproduce the results with some difficulty. The settings of parameters are underspecified or subjectively determined; the training/evaluation data are not widely available.

**Reviewer Confidence:**

5: Positive that my evaluation is correct. I read the paper very carefully and I am very familiar with related work.

---

> ### Author Rebuttal · Authors · 2023-08-29
>
> We thank you for your valuable comments and time.
>
> Responses to reasons for rejection:
>
> 1. We agree that these concepts can be explained better. However, given the space constraint of a short paper and the substantial content that we aimed to cover, we tried our best to provide the most comprehensive explanation possible. Please find below further explanations regarding the confusion that you mentioned in your comment:
>
>  * **Lines 001-006**: They express the purpose of obtaining sentence embedding representations in NLP. For a downstream task, such as sentence classification (transfer tasks), there are two ways to use BERT: (1) fine-tune the whole BERT and the MLP (classifier) head attached to it on the training data or (2) use BERT as a feature extractor by passing each sentence one time to a pre-trained BERT and get a fixed representation for it (by some pooling method), and use this representation to fine-tune only the MLP head. The second method (the purpose of sentence embedding) is favorable since it removes the need for the expensive computation of fine-tuning BERT itself.
>
> * **The proposed method not requiring training data**: We have actually mentioned the fact that our method does not require training data also in lines 059-061.
>
> * **Lines 103-106**:  They explain the different methods used for learning sentence embeddings. There are two types of models: (1) models that do not use any labeled data (2) models that use some labeled data (such as natural language inference data sets), but the sentence embeddings learned by them are used for different target data sets (such as STS data sets).
>
> We would be happy to clarify if you have any further questions.
>
> 2.
>
> * **Lines 217-224**: We have not used any mathematical symbols here. If you mean the models’ names, this is the best way we could find to define our models’ names in such a short space. However, we can add more explanation to the appendix for this if you think it is necessary.
>
> * **Line 163**: We believe that these notations are widely used for powerset and empty set in set theory. Hence, we did not explain them thoroughly. However, if you think explaining these can be beneficial, we can surely add it later.
>
> * **Lines 664-686**: We are not able to understand what mathematical symbols are not defined here. We have defined everything either in these lines or previously in lines 150-154. Please kindly refer us to the exact symbol(s) so that we can clarify.
>
> 3. We tried our best to make the algorithms as readable as possible, however, we can try and explain the algorithm even more in the text.
>
> 4. The purpose of this paper is to do sentence embedding which means a fixed vector representation should be obtained for a sentence and then be used for the downstream tasks without fine-tuning the vector again on training data. Therefore, our method should be only compared with models that do not need fine-tuning, not the fine-tuned models. Thus, by state-of-the-art, we mean the state-of-the-art result for these data sets which are designed for the task of sentence embedding.
>
> Responses to questions:
>
> We did not experiment with all the models and data sets as we saw in our early experiments that the max pooling results are not as good as mean pooling’s. The following table shows the results for using max pooling function for certain models and STS data sets that we have. As you can see, our method still significantly outperforms the baselines; however, the overall Spearman correlations of max-pooling are lower than mean-pooling's.
>
>  | Model| STS16 | STSB | SICK    | Avg
> | :---        |    :----:   |          ---: |           ---: |          ---: |
> | Roberta-B-last      | 53.54        | 50.22   | 59.54  | 54.43
> | Roberta-B-LC      | 62.41        | 60.19  | 62.33  | 61.64
> |BERT-base-uncased-CLS Pooler      | 27.52        | 11.72   | 39.23  | 26.16
> | BERT-base-uncased-last       | 59.7        | 54.1 | 56.89   | 56.9
> | BERT-base-uncased-LC       | 67.63        | 65.11   | 59.92  | 64.22
>
> 2. Please note that since all the base models (such as BERT, RoBERTa, SBERT, etc.) have the exact same architecture and the same number of weights, obtaining the tensor of all layer and token vectors by any one of them takes the same amount of time. It also takes the same amount of time to iterate through all of the layer combinations with the same method for all of these models. Hence, the results that we have reported for BERT-base-uncased are valid for any other base model.
>
> 3. All the BERT-based models have an MLP head attached to them. This head is attached to the output of the first token in the BERT’s last layer. BERT originally used this head to perform a binary task of next sentence prediction. However, as seen in the results and explained in lines 155-160, using the output of this MLP head leads to poor performance for the task of sentence embedding. However, SimCSE trains this head towards learning sentence embeddings. Therefore, this head holds important information for our task of sentence embedding. If we just use a mean pooling of the layers in SimCSE, as we are doing with all the other models, we are ignoring this important information. Thus, as explained in line 171-175, we propose an extension to our method. As before, we try to get a sentence embedding by mean-pooling from a particular set of layers, D. However, after obtaining p(H,D), we then pass p(H,D) to the MLP head of SimCSE, and use its outputs as p’(H,d) which as our new and final sentence embedding.

---

### Official Review · Reviewer_YvLi · 2023-08-03

**Soundness:** 2

**Excitement:**

2: Mediocre: This paper makes marginal contributions (vs non-contemporaneous work), so I would rather not see it in the conference.

**Missing References:**

[1] Neelakantan A, Xu T, Puri R, et al. Text and code embeddings by contrastive pre-training[J]. arXiv preprint arXiv:2201.10005, 2022.

**Paper Topic And Main Contributions:**

This paper proposes a new method for obtaining sentence embeddings by combining certain layers of a BERT-based model. The authors address the problem of improving the quality of sentence embeddings, which are widely used in various natural language processing tasks such as text classification, information retrieval, and machine translation.

**Reasons To Accept:**

1. A new method for obtaining sentence embeddings by combining certain layers of a BERT-based model, which outperforms the baseline BERT on various tasks and data sets, achieving state-of-the-art results.
2. An analysis of the effect of different layers and combinations of layers on the quality of sentence embeddings, which provides insights into the inner workings of the BERT model and how it can be fine-tuned for specific tasks.
3. A comparison of the proposed method with other state-of-the-art methods for obtaining sentence embeddings, which shows that the proposed method is competitive and often outperforms them.


**Reasons To Reject:**

1. The proposed method is based on a relatively simple modification of the BERT model, which may not be novel or significant enough to warrant publication in a top-tier conference like EMNLP.
2. The evaluation of the proposed method is limited to a few standard tasks and data sets, which may not be sufficient to demonstrate its effectiveness in real-world scenarios or to compare it with other state-of-the-art methods, mostly from 2022 and 2023.
3. The paper does not provide a detailed analysis of the interpretability of the proposed method, which could limit its usefulness in some applications where interpretability is important.
4. The paper does not compare the proposed method with other recent methods that have achieved state-of-the-art results on similar tasks, which could provide a more comprehensive evaluation of the proposed method's performance.


**Reproducibility:**

4: Could mostly reproduce the results, but there may be some variation because of sample variance or minor variations in their interpretation of the protocol or method.

**Reviewer Confidence:**

3: Pretty sure, but there's a chance I missed something. Although I have a good feel for this area in general, I did not carefully check the paper's details, e.g., the math, experimental design, or novelty.

---

> ### Author Rebuttal · Authors · 2023-08-29
>
> We thank you for your valuable comments and time.
>
> Responses to reasons for rejection:
> 1. We agree that our proposed method is simple. However, while the idea itself might not be intricate, we argue that its practical implications and advantages are indeed significant. Our proposed method offers the following advantages:
>
> * **Effective and Easily Adaptable**: It is effective and easy to adapt to existing methods which can potentially open up new avenues for further research and application.
>
> * **No Fine-tuning Required**: It does not require fine-tuning.
>
> * **Proposed Optimized Algorithm**: We introduce a novel algorithm which speeds up finding the best layer combination by a factor of 189 times for base models.
>
> * **Extensive Array of Experimentations**: We show the effectiveness of our proposed method on two different sentence embedding tasks and 15 different datasets. We conducted an extensive array of experimentations across many different datasets and baseline models, establishing a valuable reference framework for future endeavors in this field.
>
> * **State-of-the-art Result for Transfer Tasks**: We also show that our method achieves state-of-the-art result for transfer tasks while improving over a large set of BERT-based models for STS tasks.
>
> * **Good Performance in Limited Data Settings**: We show that our method outperforms the baselines even in limited data settings.
>
> * **Comprehensive Ablation Studies**: We also conducted different ablation studies which provide empirical evidence of the impact of different layer combinations on various tasks.
>
> We understand the rigorous standards of EMNLP and are committed to ensuring that our work meets the criteria for acceptance. We believe that the novelty of our work along with the above-mentioned contributions is reasonable for being accepted as a short paper in EMNLP.
>
> 2. This paper exclusively focuses on the area of sentence embedding utilizing an extensive array of datasets most widely used by other researchers for sentence embedding tasks. We intend to apply our method to additional models and other tasks; however, we believe this falls beyond of the scope of a short paper.
>
> 3. We recognize the importance of analyzing the interpretability of the proposed method. However, our current paper primarily focuses on establishing the effectiveness and performance of our proposed method, not interpretability.  We do not claim that our method can be applied to all the tasks; however, we do intend to thoroughly address the interpretability aspect and potential applicability of our method in future work.
>
> 4. We have applied our method to a variety of different BERT-based models that include both supervised (SBERT and SimCSE) and unsupervised models, showing the effectiveness of our method in different scenarios. We can extend our work to newer models for future work.

---

### Official Review · Reviewer_mPVd · 2023-08-07

**Soundness:** 4

**Excitement:**

4: Strong: This paper deepens the understanding of some phenomenon or lowers the barriers to an existing research direction.

**Paper Topic And Main Contributions:**

the paper proposes to use average pooling across different combinations of transformer layers instead of just using it on the last layer. they tune the choices of layers using labeled data from downstream tasks. in experiments, they benchmark this approach on senteval toolkit across many baseline pretrained transformer models, finding that their approach consistently improve the performance significantly.

**Questions For The Authors:**

- in lines 213-215, what do you mean by the training set was not used by your methods? Did you tune your layer combinations on the validation sets?

**Reasons To Accept:**

- the approach is simple to understand and works quite well
- the experiments are thorough

**Reasons To Reject:**

- the approach seems to be a lot more computationally expensive than the baseline as each layer combination will require re-trainining the top classifier. it's unclear how easy the approach can generalize to other tasks, but this is probably beyond the scope of a short paper

**Reproducibility:**

3: Could reproduce the results with some difficulty. The settings of parameters are underspecified or subjectively determined; the training/evaluation data are not widely available.

**Reviewer Confidence:**

3: Pretty sure, but there's a chance I missed something. Although I have a good feel for this area in general, I did not carefully check the paper's details, e.g., the math, experimental design, or novelty.

---

> ### Author Rebuttal · Authors · 2023-08-29
>
> We thank you for your valuable comments and time.
>
> - We agree that this is more computationally expensive. However, please kindly note that this is true only for the transfer tasks and not for STS tasks because there is no training required for STS.
>
> Questions:
> - We mean that we do not use the training data to obtain sentence embeddings in our models. We obtain the best layer combination from the validation data set. The training data is just used for training a logistic regressor using the feature vector obtained for each of the sentences by our model.

---

### Meta-Review · Area_Chair_LJMq · 2023-09-12

**Recommendation:** 3

**Metareview:**

The paper proposes a very simple approach for obtaining sentence embeddings: combine representations from different layers (and not necessarily from the last layer only, which is the common procedure). Consistent improvements are reported across many datasets and tasks. There is a consensus among the reviewers on the simplicity and effectiveness of the approach. There were some concerns on the writing of the paper and computational complexity; the authors are expected to work on these. The issues raised by the reviewer YvLi on the limited experiments and interpretability studies are not valid, given the focused scope of this short paper. The SentEval benchmark (8 datasets) is a standard benchmark usually used in the domain. The authors also report results on the STS benchmark (6+ datasets). Interpretability analysis is indeed out of scope.

---

### Decision · Program_Chairs · 2023-10-07

**Decision:**

Accept-Findings

**Comment:**

The paper proposes a very simple approach for obtaining sentence embeddings: combine representations from different layers (and not necessarily from the last layer only, which is the common procedure). Consistent improvements are reported across many datasets and tasks. There is a consensus among the reviewers on the simplicity and effectiveness of the approach. There were some concerns on the writing of the paper and computational complexity; the authors are expected to work on these. The issues raised by the reviewer YvLi on the limited experiments and interpretability studies are not valid, given the focused scope of this short paper. The SentEval benchmark (8 datasets) is a standard benchmark usually used in the domain. The authors also report results on the STS benchmark (6+ datasets). Interpretability analysis is indeed out of scope.